# Extraction, Identification, and Quantification of Polyphenols from the *Theobroma cacao* L. Fruit: Yield vs. Environmental Friendliness

**DOI:** 10.3390/foods13152397

**Published:** 2024-07-29

**Authors:** Juan Manuel Silva, Fernanda Peyronel, Yinan Huang, Carlos Eugenio Boschetti, Maria G. Corradini

**Affiliations:** 1Institute of Biotechnological and Chemical Processes (IPROByQ-CONICET), National University of Rosario (UNR), Rosario 2000, SF, Argentina; js.juanmanuelsilva@gmail.com (J.M.S.); ceboschetti@gmail.com (C.E.B.); 2Food Science Department, University of Guelph, Guelph, ON N1G 2W1, Canada; yh699@cornell.edu; 3Arrell Food Institute, University of Guelph, Guelph, ON N1G 2W1, Canada

**Keywords:** *Theobroma cacao* L., cacao, cocoa, polyphenols, extraction, analytical methods, green chemistry, sustainability, yield, penalty points

## Abstract

The cacao fruit is a rich source of polyphenols, including flavonoids and phenolic acids, which possess significant health benefits. The accurate identification and quantification of these bioactive compounds extracted from different parts of the cacao fruit, such as pods, beans, nibs, and cacao shells, require specific treatment conditions and analytical techniques. This review presents a comprehensive comparison of extraction processes and analytical techniques used to identify and quantify polyphenols from various parts of the cacao fruit. Additionally, it highlights the environmental impact of these methods, exploring the challenges and opportunities in selecting and utilizing extraction, analytical, and impact assessment techniques, while considering polyphenols’ yield. The review aims to provide a thorough overview of the current knowledge that can guide future decisions for those seeking to obtain polyphenols from different parts of the cacao fruit.

## 1. Introduction

Cocoa products attract considerable scientific interest due to the health claims derived from their high content of bioactive compounds, notably phenolic compounds, praised for their antioxidant capacity. The cacao pod and its parts, i.e., cacao pod husks (CPHs), nibs, and cacao bean shells, have surfaced as promising sources of bioactive compounds.

The content and composition of polyphenols of the cacao pod depend not only on factors directly affecting the feedstock, such as tree genotype, growing conditions, and the ripeness degree of cacao pods at the time of harvesting, but also on the processing methods employed at both the farm and the industry level. The quantification of polyphenols is also contingent upon the methodology used for their efficient extraction from the feedstock and the subsequent analytical techniques utilized for measurement.

The *Theobroma cacao* L. tree, indigenous to the Amazonian rainforest, is cultivated globally, particularly in regions within 20 degrees north and south of the Equator, with significant production occurring in West Africa, Southeast Asia, and Central and South America. The fruit consists of an outer cacao pod husk (CPH) that surrounds the seeds (cacao beans), intermingled with a white mucilage. Cacao beans comprise a cotyledon and a coat or cacao shell (Figure 1). Conventionally, “cacao” refers to the unprocessed parts of the *Theobroma cacao* L. fruit, while “cocoa” typically denotes processed parts, a distinction adhered to herein.

Processing of cocoa beans at the farm typically involves discarding the CPHs and fermenting the mucilage together with the beans, a step crucial for flavor, color, and polyphenol content modulation. Fermentation causes cell destruction in the cotyledon, facilitating the release of polyphenols from storage cells [1]. Once the polyphenols are not protected by the cell, they are exposed to pH and temperature changes, as well as many enzymes present in the cotyledon. The total polyphenol content in fermented and dried whole cocoa beans has been reported in the range of 12–18% (d.w.) [2,3]. Several studies have shown that fermentation notably decreases polyphenol levels when compared with non-fermented beans [4,5,6]. Certain polyphenols may increase post-fermentation, influenced by genetic factors such as anatomical features of the beans [7]. Fermentation is immediately followed by sun-drying the beans before they are stored for transportation to the processing industries. 

Drying and storage of the beans have a minor influence on the polyphenol content. However, the roasting process of the cacao beans in the industry, which is the initial step in the manufacture of chocolate, significantly impacts the polyphenol levels. The thermolabile molecular structures of polyphenols are affected by high and prolonged roasting temperatures. Roasting temperatures ranging from 70 to 150 °C for 15 to 45 min have been shown to reduce total polyphenol content in cacao feedstock [4,8,9]. Milder roasting treatments (~100 °C) decrease flavanol content by 18%, while more intense roasting (~150 °C) results in a 50% reduction compared to non-roasted beans [4]. Moreover, bean provenance has also been correlated to polyphenol thermal stability. For example, flavanols in cocoa beans from the Ivory Coast were less affected by roasting than those from Java [4]. Furthermore, it was observed that roasting at particular temperatures causes certain polyphenols to increase at the expense of others [10]. Migration of polyphenols from the bean interior to its shell occurs during roasting, rendering the shell a potential source of ingredients for food applications.

This review used the Eco-Scale penalty point system [11] to compare protocols for the extraction, identification, and quantification of polyphenolic compounds across different cacao fruit feedstocks. This penalty point system was employed to determine the potential impact the method and the analytical techniques have on human health as well as the environment. This review shows that polyphenol yield is highly dependent on the methods used for extraction, identification, and quantification, which also play a role in the sustainability of a methodology.

Readers can utilize this information to identify environmentally friendly extraction procedures and analytical quantification methods for characterizing polyphenols in cacao feedstocks. Additionally, if the reader is interested in implementing an appropriate scale-up procedure, this review should offer valuable insights into the efficacy and sustainability of industrial extractions and monitoring processes. 

## 2. Molecular Structure of the Polyphenols in Cacao Feedstock

The health benefits of polyphenols are primarily attributed to their molecular structure and degree of polymerization [12,13,14]. The majority of cocoa polyphenols (>90%) occur as polymers, with monomers only accounting for 5–10% [15]. 

Polyphenols are classified into different groups according to their molecular structure. The number of aromatic rings and the structural elements that link these rings together determine which category a polyphenol belongs to. Figure 2 shows the names of the main polyphenols present in cacao, while Appendix A presents details of their chemical structures. 

Flavonoids (Appendix A), which include flavanols (also called flava-3-ols), anthocyanins (Appendix A), and flavonols (Appendix A), are the largest group of phenolic compounds in cacao and cocoa byproducts [16]. Flavanols are the principal polyphenols in cocoa beans, followed by anthocyanins and flavonols like tannins [17,18]. Among the non-flavonoid compounds in cacao, phenolic acids are the most commonly found [19,20].

Catechin (trans-isomer) and epicatechin (cis-isomer) are the most abundant flavanols. Both of them have two stereo-isomers, i.e., (+)-catechin, (−)-catechin, (+)-epicatechin, and (−)-epicatechin (Appendix A). The most important are (−)-epicatechin, which constitutes 30–40% of total polyphenols in cacao, and (+)-catechin, in the form of monomers or oligomers such as proanthocyanidins (Appendix A) [2,21,22]. Proanthocyanidins are classified according to their degree of polymerization (DP); DP equal to 1 refers to monomers, 2–10 are considered oligomers, and >10 are polymers [23]. Procyanidins are classified into types A and B based on their stereo-configuration and the link between monomers. The B-type is the predominant form in cocoa products [16,24], while gallocatechin and epigallocatechin are present in smaller amounts [25,26,27]. For instance, the epicatechin content of cocoa shells varies from 0.31–2.24 mg g^−1^ across 12 different bean types [28].

Anthocyanins are pigments known for their antioxidant and antimicrobial activities [16,20] (Appendix A). This group of polyphenols constitutes ~4% of the total polyphenols in cocoa products. However, the fermentation process contributes to the degradation of some of them, e.g., cyanidin-3-galactoside and cyanidin-3-arabinoside [2,29]. Among the flavonols, which are glycosylated compounds similar to the anthocyanins, quercetin-O-glycosides, such as quercetin-3-O-galactoside, quercetin-3-O-arabinoside, and quercetin-3-O-glucoside, have been identified in cocoa beans [30,31].

Non-flavonoid polyphenols are present in small quantities in cacao beans [32]. Phenolic acids (Appendix A) such as N-caffeoyl-3-O-hydroxytyrosine (clovamide), N-p-coumaroyl-tyrosine (deoxyclovamide), cinnamoyl-L-aspartic acid, and caffeoyl-glutamic acid (see Appendix A) have been reported in the different cacao fruit parts. These compounds, as well as procyanidins, contribute to the flavor of both unfermented cacao beans and roasted cocoa nibs [18].

## 3. Green Chemistry and the Eco-Scale

The purpose of applying green chemistry principles to extraction and analytical techniques is to eliminate, as much as possible, solvents, processes, and byproducts that can negatively impact human health and the environment [33].

The Eco-Scale [11,34] evaluates a process’s overall impact based on the potential pollution of reagents/solvents and equipment energy consumption using penalty points (PP). The higher the total PP, the more harmful and less green the process. In this review, the PP associated with reported polyphenol extraction and identification/quantification methods, considering the amount of reagent or solvent used and equipment energy consumption, are reported and compared. PP are assigned based on the Globally Harmonized System of Classification and Labeling of Chemicals (GHS) pictograms and signal words. The Eco-Scale also considers waste generated, and its post-treatment, which contributes to the total PP for the method. The Eco-Scale’s merit is that the whole process’s environmental impact becomes quantifiable, allowing easy comparison between different methods.

This review reports and summarizes the penalty points (PP) for all the surveyed extraction and identification steps for polyphenols from the cacao feedstocks. PPs calculated for (1) pre-treatment and extraction steps, and (2) identification or quantification steps are presented in independent tables. Equation (1) was utilized to estimate the total PP (PP_T_) for each step [11,34]:(1)$${PP}_{T}={PP}_{C}+{PP}_{E}$$
where *PP_C_* are the penalty points accrued from the amount and type of chemicals used, and *PP_E_* are those contributed from the consumption of energy.

Table 1 shows the criteria used to determine the PPs from the chemicals used and the consumed energy during each step [11,34].

The PP associated with a particular chemical (i.e., reagent or solvent) employed during a selected process (pre-treatment, extraction, identification, and quantification) were calculated by multiplying the amount used (PP = 0, 1, 2, or 3) by the PP due to the reagent/solvent risk. The reagent/solvent risk was calculated by multiplying the PP associated with the number of pictograms on the chemical label (PP = 0, 1, 2, or 3) by the PP related to the type of signal words displayed on the label (PP = 0, 1, or 2), as shown in Table 1. The pictograms and signal words should comply with the Globally Harmonized System of Classification and Labeling of Chemicals (GHS). The assignment of PP was carried out on a per-pictogram basis, meaning that if two pictograms were present, two PP were assigned. Appendix A provides a list of common reagents and solvents often reported in polyphenol extractions and identification, along with their corresponding reagent/solvent risk PP. This information is helpful in selecting greener alternatives to process and characterize cacao and cocoa products.

The primary criterion used to determine the penalty points associated with the use of equipment, as indicated in Table 1, is energy consumption. PP and energy consumption of equipment commonly used for the extraction and identification of polyphenols are provided in Appendix A.

## 4. Polyphenols Extraction Methods

Separating polyphenols from vegetal matrices typically involves a sequential process comprising pre-treatment, extraction or soaking, and sample concentration. Pre-treatment involves preparatory techniques aimed at enhancing the polyphenol yield before extraction/soaking of the feedstock. In this review, the term feedstock encompasses all parts of the cacao fruit, including CPHs, beans, cacao shells, and nibs. During extraction or soaking, a solvent is utilized to facilitate the migration of polyphenols from the solid matrix to the liquid phase. Solid residues are then separated from the solvent-containing polyphenol extract through filtration or centrifugation. Sample concentration is frequently attained through solvent evaporation, while commercialization often necessitates extract purification or isolation. However, this review focused primarily on comparing extraction and identification/quantification procedures; therefore, sample concentration and purification were excluded from the analysis.

### 4.1. Pre-Treatment of the Samples

The selection of suitable pre-treatment steps is critical when dealing with labile compounds such as polyphenols, which degrade, oxidize, polymerize, or form complexes over time. Several pre-treatments used to increase polyphenols’ extraction yield from cacao feedstocks have been reported in the literature. These pre-treatments can be categorized based on their primary mechanism into: (i) increasing the surface area exposed to the solvent, (ii) water removal, (iii) lipid removal, and (iv) loosening intracellular compounds in the feedstock.

Size reduction of the feedstock, for example, by milling, increases the surface area exposed to the solvent, thus enhancing polyphenol yields. Milling can be achieved using a centrifugal or a cutting mill [35,36,37] or by crushing cacao parts with a knife mill [4,37]. Sieving after milling is typically performed to obtain a monodisperse particle-size distribution ranging from 0.5 mm to 1 mm [38,39,40]. However, friction during milling might increase the temperature, which can promote the complexation of procyanidins with matrix components and hinder their subsequent extraction. Drying the samples before milling can prevent caking, facilitate the breaking of the cell structure [41,42,43], and preclude undesirable enzymatic reactions [44].

Drying is necessary for raw CPHs or cacao beans due to their high water content. Fermented cocoa beans can be sun-dried, eliminating the need for additional drying. Freeze- and air-drying are the preferred methods to reduce water content, especially for raw beans or pod husks [45,46,47]. Freeze-drying is less disruptive than air-drying for heat-labile compounds like polyphenols, and air-drying requires longer times and higher temperatures (>100 °C), leading to the degradation of some phenolic compounds. Microwave-, freeze-, and hot-air-drying were compared for CPH [48], with hot-air-drying at 60 °C for 24 h resulting in a reduction of ~60% of total polyphenols, flavanols, and flavonoids compared to both freeze- and microwave-drying.

The defatting process involves soaking the milled–dried solid material in solvent multiple times, leading to higher yields of polyphenols, the targeted analytes [39,42,47]. Commonly used solvents are hexane and petroleum ether, although less common ones include heptane and dichloromethane [41,49,50]. Soxhlet defatting results in greater fat extraction efficiency but requires longer running times and higher energy consumption [26,46]. In contrast, mechanical defatting, such as pressing, does not use a solvent and is more environmentally friendly. Kobori et al. [51] reported that mechanical defatting of cocoa powder improved total polyphenol and procyanidin content by 14% and 19%, respectively, compared to extractions of non-defatted cocoa powder.

Pulsed electric field (PEF) pre-treatments release intracellular compounds, like polyphenols, by permeabilizing the cell membranes. Since PEF is a non-thermal treatment, it does not affect thermolabile compounds like polyphenols. Barbosa-Pereira et al. [28] applied PEF to pre-treat cocoa bean shells immersed in water. These authors showed that by adequately selecting the parameters of the PEF (e.g., duration of pulse and strength), the extractability of polyphenols increased by about 30%.

Table 2 summarizes the various pre-treatment methods utilized for cacao feedstocks, including pod husks, beans, nibs, and shells, along with their respective states (e.g., raw, sun-dried).

The listed penalty points (PP) in Table 2 are the sum of those associated with each pre-treatment and extraction method, calculated using Equation (1) and employing the techniques documented in the scientific literature. The environmental impact assessed using the Eco-Scale based on the information in Table 2 revealed that milling is an efficient and speedy process, with energy consumption penalties from 0 to 2 PP, depending on the equipment employed (Appendix A). Although freeze- and oven-drying are straightforward steps, they can result in 2 to 4 PPs due to their energy consumption. Defatting, on the other hand, requires solvents such as hexane or petroleum ether, resulting in a substantial increase in PPs (8 to 12 PP) when using between 10 and 100 mL. Pulsed electric field (PEF) is an eco-friendly pre-treatment when using water as the soaking medium, with energy consumption being the only contributor to the PPs, resulting in a single PP [28]. However, substituting methanol for water can increase the PPs to ~12 when using 10 to 100 mL of solvent (Appendix A). Lower penalty points are always preferred as they indicate a more efficient and cost-effective method. The selection of appropriate pre-treatment methods is crucial as it can significantly impact the final penalty point counts. Therefore, careful consideration and evaluation should be given when deciding which pre-treatment method(s) to utilize in order to retain the efficiency of the extraction with minimal environmental impact.

### 4.2. Extraction Procedures

The extraction process aims to achieve the highest possible yield of polyphenols by optimizing the processing conditions. Solid–liquid extraction methods have traditionally been used to obtain polyphenols from cacao feedstock. These methods involve immersing the material in a chosen solvent for a specific time. The yield can be increased by facilitating mass transfer through forced convection (i.e., stirring), increasing the temperature, or applying ultrasound during the extraction process. On the other hand, supercritical fluid extraction is gaining popularity due to its high yields and its classification as a green technique, owing to the utilization and recycling of a supercritical fluid. Table 2 compares the conditions reported in the literature for extracting polyphenols from various cacao feedstocks, along with their corresponding penalty points (PP).

Solid–liquid extraction techniques that use solvents other than water are categorized as non-green techniques. A “greener method” incurs fewer PPs compared to other methods. Ideally, a green technique would not incur any PP, but such a technique has not been developed yet. Solvent selection during extraction plays a crucial role in determining the PPs associated with the technique. Both methanol and ethanol have higher PP values than acetone (6 vs. 4 PP). In contrast, water incurs no penalty points. The pH of the solvents can also contribute to PP, depending on the type and quantity of reagent used. For example, formic acid, acetic acid, and hydrochloric acid are each assigned 4 PP. The volume of solvent used in the extraction process can also significantly impact the reagent PP value. Increasing the solvent volume from 1 to 10 mL can double the latter PP value, as presented in Table 1.

To select an appropriate extraction method, both the extraction yield and the associated penalty points should be considered. Focusing solely on the latter can lead to misleading choices. In some cases, a method with higher PPs may still be more efficient in terms of yield than a method with lower PP. For instance, Nsor-Atindana et al. [59] extracted total polyphenols from cocoa bean shells using water (0 PP), acetone (4 PP), ethanol (6 PP), and methanol (6 PP), obtaining yields of 17.2, 42, 23, and 25 mg eq. g^−1^ (d.w.), respectively. In this case, selecting a method based solely on penalty points may lead to a lower yield since acetone, which has a moderate PP contribution, allowed for doubling the extraction efficacy. Hernández-Hernández et al. [26] compared the effectiveness of five solvent combinations, including water, acidified water, methanol–water 80:20 (*v*/*v*), methanol–acidified water, and ethanol–acidified-water, in combination with acetone–water 70:30 (*v*/*v*), to extract total polyphenols from raw cacao nibs. The reagent risk penalty points for all the listed solvents were 0, 4, 6, 11, and 14, respectively, and the corresponding yields were 5.7, 9.4, 14.6, 20.4, and 49.5 mg eq. g^−1^ (d.w.). Therefore, a balance between penalty points and yield should be considered when selecting an extraction method, as solely focusing on penalty points can result in an ineffective or non-profitable extraction with low yields or a high-yield method that generates waste requiring additional remedial steps.

Due to its environmentally friendly status, water has been extensively explored as a viable solvent for extracting polyphenols. Camu et al. [35] and Manzano et al. [45] extracted defatted cocoa nibs and cocoa shells, respectively, with water, obtaining modest yields of ~0.025 and 6 mg eq. g^−1^ d.w., respectively. Using water as a solvent is highly desirable when pursuing a greener methodology; however, this often results in lower extraction yields. Consequently, when transitioning to greener alternatives (e.g., switching from methanol to water as an extraction solvent), the steps to increase yield should be considered [26]. The use of physical or mechanical treatments can increase the efficacy of water during the extraction process, as long as the selected treatments result in lower penalty points compared to more efficient solvents.

Among these treatments, ultrasound and pressurized assisted extractions can help release polyphenols from the feedstock into the solvent. Ultrasound-assisted extraction (UAE) applies ultrasound while the feedstock is being soaked in the solvent. It is recognized as a green technology due to its low energy consumption, i.e., ≤1.5 kWh (see Appendix A), and lack of requirements for high temperature and additional solvents [37,41,46,47,49,54,60,61]. Using ultrasound increases polyphenol extraction efficiencies by ~30% compared to soaking alone [54]. Although UAE can be performed at room temperature, increasing the temperature during the water-soaking step can increase the total polyphenol content by 15% [46].

Pressurized liquid extraction (PLE) is a technique that uses elevated temperatures and pressure to rapidly extract compounds from a sample. The feedstock and solvent are placed in a pressurized vessel, facilitating extraction. When water is used as the extraction solvent, the process is referred to as pressurized hot water extraction (PHWE), sub-critical water extraction, or superheated water extraction [62]. PLE is a low-cost method that achieves high yields without the need for harmful solvents or pre-treatment steps, such as defatting. Plaza et al. [49] compared the efficiency of PHWE at 125 °C with UAE at 30 °C for extracting polyphenols from cocoa nibs, beans, and chocolate. The PHWE method resulted in a ~6-fold increase in total polyphenol yield compared to UAE, particularly for procyanidins in nibs and beans. However, there was no significant difference in yield between the two methods for chocolate samples.

Supercritical fluid extraction (SFE) is a promising alternative to conventional liquid–solid techniques that is gaining increasing attention due to its potential as a green methodology. SFE utilizes a solvent at supercritical conditions, where its pressure and temperature are modulated to achieve the supercritical state, resulting in the separation of the extractant from the feedstock. The use of a supercritical fluid, characterized by gas-like viscosity and diffusivity, facilitates the mass transfer of analytes from the matrix to the fluid and its circulation within confined spaces, combined with a liquid-like density that imparts high solvation power to the fluid [63]. The most commonly used supercritical fluid is CO_2_, which presents a low reagent risk (2 PP—See Appendix A). SFE also offers the advantage of CO_2_ recyclability, making it an eco-friendly method that eliminates the need for waste treatment and minimizes environmental contamination. Other reagents, such as water (0 PP) and ethanol (5 PP), can be added as co-solvents. SFE has demonstrated consistent and superior extraction yields of polyphenols in comparison to other techniques. For instance, the SFE with CO_2_ of cocoa beans has been reported to yield 43.3–64.2 mg eq. g^−1^ (d.w.) [59]. In a study by Mazzutti et al. [58], SFE with CO_2_ and hexane extraction were compared in terms of total extracted polyphenols from fermented and roasted cocoa shells. Although the yields were similar (4.0 vs. 4.5 mg eq. g^−1^ d.w.), the reagent risk associated with hexane (8 PP) outweighed the low risk of CO_2_ (2 PP), which underscores the growing popularity of SFE as an extraction alternative.

Table 2 summarizes not only the type and conditions of the pre-treatments but also those of the extraction methods reported. The utilization of solvents such as methanol, acetone, or acetic acid in extraction procedures can result in elevated PP values when compared to solvents like water or ethanol. Even when using 10–100 mL of such solvents, an increase of 8 to 12 PP can be observed. In addition, the use of sophisticated equipment like rotary evaporators, pressurized vessels, or freeze-dryers incurs a penalty of 2 PP each due to their reagent requirements, which, coupled with their energy consumption, can increase the PP to 3 or more. However, it is crucial to note that selecting an extraction method should not solely be based on the penalty points incurred, as higher PP values may not necessarily reflect higher efficiency in terms of polyphenol yield, as has been already mentioned for the pre-treatments and will be further discussed in this review.

### 4.3. Optimization of the Extraction Procedure

The quest for environmentally friendly chemical approaches has led to the exploration of two distinct pathways, which are not mutually exclusive. The first approach involves replacing, recycling, or eliminating harmful solvents, reagents, or energy-intensive processes. The second approach optimizes outcomes such as extraction yields by systematically exploring processing conditions using an adequate design-of-experiments (DOE) approach. This strategy can lead to a method that is more efficient and eco-friendlier. The DOE approach consists of two main steps: factor screening and optimization. The response surface methodology (RSM) is commonly used to optimize the factors and their values to achieve the best possible outcome. This optimization technique has been successfully applied to the extraction of polyphenols from cocoa products by identifying suitable pre-treatment conditions [28], selecting appropriate temperature, time, and reagent concentrations during traditional extractions [43,46], and identifying the most effective temperature, pressure, and reagent concentrations in an SFE-CO_2_ process [49]. Although significant improvements have been achieved through appropriate DOE and optimization techniques, most previous studies selected extraction conditions based solely on prior reported work, without considering sustainability performance parameters (PP). The integration of PP in the optimization process remains largely unexplored but holds great potential for further improving the eco-friendliness of chemical processes.

## 5. Polyphenol Identification and Quantification Procedures

Accurately quantifying phenolic compounds is essential for comparing the yields obtained by various extraction methods and determining their selectivity towards different phenolic components. The quantification techniques can be broadly categorized into two groups: spectrophotometric- and chromatographic-based methods. These techniques enable researchers to precisely measure the concentration of phenolic compounds in a sample, thereby facilitating the identification and characterization of these bioactive compounds.

### 5.1. Spectrophotometric Methods

UV–Vis absorbance spectroscopy is widely utilized for the identification and quantification of polyphenols. This technique is based on the capability of certain compounds, referred to as chromophores, to absorb light. In the absence of natural chromophores in the sample, it is necessary to prepare the sample to elicit a detectable color response by promoting a reaction between the compound of interest, such as a polyphenol, and a suitable chromophore. The observed absorbance is then compared to a calibration curve obtained using a standard reagent, which is used to determine the quantity of polyphenols present in the sample. The polyphenol concentration is usually expressed as milligrams of calibration standard equivalents per gram of sample, usually based on dry weight. Only a small amount of the standard is required for this analysis. However, the use of specific reagents, such as gallic acid and epicatechin, incurs penalty points related to their reagent risk PP, as outlined in the Appendix A.

Several assays can produce a color cue from polyphenols, including Folin–Ciocalteu (F–C), vanillin, aluminum chloride, 4-(dimethylamino) cinnamaldehyde (DMAC), and acid butanol. However, the F–C assay is the most used due to its simplicity and comprehensive results. Nevertheless, the assay’s limited specificity toward polyphenols is a significant drawback since it reacts with several oxidation substrates in the sample, including but not limited to phenols [64,65]. Upon reacting with the F–C reagent in a Na_2_CO_3_ solution, phenolic compounds form blue complexes, which absorb at 740–765 nm [66]. The kinetics and efficacy of the reaction depend on factors such as reagent volume and concentration, temperature, and environmental light exposure. The total polyphenol content is determined from a standard curve of gallic acid, catechin, or (−)-epicatechin. Ramirez-Sanchez et al. [67] reported that (−)-epicatechin demonstrated more sensitivity than gallic acid and is preferred as a standard since it is present in all cacao and cocoa products. The penalty points associated with this method on the Eco-Scale are relatively low and typically amount to <5 PP, with 1 PP attributed to the amount of reagents (<10 mL), up to 3 PP for reagent risk, and 1 PP for the use of a spectrophotometer.

The quantification of total flavonoids is commonly carried out through the aluminum chloride (AlCl_3_) colorimetric assay [68]. The extract is typically treated with a solution of sodium nitrite and sodium hydroxide to enhance the method’s selectivity, followed by the addition of an AlCl_3_ solution [69]. The presence of NaNO_2_ in an alkaline medium allows the determination of rutin, luteolin, and catechins. Conversely, the absence of NaNO_2_ only permits the quantification of flavonols and luteolin. The absorbance is measured at 500–510 nm, and the reaction time is approximately 15–30 min [70]. Standards such as catechin [28], rutin [40,70], (−)-epicatechin [38], and quercetin [71] are commonly employed. The addition of NaNO_2_ increases the selectivity of the assay, but also contributes to 6 PPs on the Eco-Scale, resulting in a total of 16 PPs. Thus, it is recommended to avoid using NaNO_2_ unless its selectivity is required for the analysis.

Total flavanols can be determined using the 4-(dimethylamino) cinnamaldehyde (DMAC) method. This method employs a DMAC reagent that specifically reacts with (−)-epicatechin, (+)-catechin, epigallocatechin, and gallocatechin, producing a green coloration that can be measured at 640 nm [72]. A DMAC in HCl/ethanol solution is utilized, with 10% *w*/*v* of HCl being preferred to achieve shorter reaction times. Replacing ethanol with water as the solvent slows the reaction with DMAC, and thus, it is not recommended [48]. Epicatechin is commonly used as the calibration standard [48,57]. However, it is worth noting that the DMAC method has a relatively high score of 11 PP.

The vanillin assay is a method used to quantify flavonols instead of total polyphenols. Flavonols react with vanillin (2.4% *w*/*v*) and methanol under acidic conditions (HCl), and the intensity of the resulting red coloration is measured at a wavelength of 500 nm [73,74]. The absorbance of the derivatized sample is compared to a calibration curve typically prepared with catechin. To increase the stability of the analyte/vanillin complex, the reaction should be protected from light. The reaction should be carried out at a constant temperature of 25 °C for 15 min to minimize variability. Still, this step is not associated with high energy consumption and carries no penalty points. The sensitivity of the method is improved when the hydrochloric acid (HCl) is diluted with methanol instead of water at a ratio of 30% *v*/*v* [75]. However, due to the need to optimize the method for better sensitivity, particularly the additional risk associated with the reagents, this method carries 11 PP. Replacing hydrochloric acid with sulphuric acid [42] reduces the PP from 11 to 9 and was found to render a better catalyst [76]. Any reduction in penalty points is considered an improvement, even if it is only a reduction of 2 PP.

When it is necessary to determine the total proanthocyanidin content of an extract, the acid butanol assay is employed. This assay involves dissolving a Fe(III) salt, such as NH_4_Fe(SO_4_)_2_ x12 H_2_O, in a concentrated HCl–butanol solution (usually in a ratio of 1:4 or 1:6) [23,55]. Under acidic conditions, leucoanthocyanins are depolymerized, and the corresponding anthocyanidins’ absorbance is measured at 550 nm [77,78]. Cyanidin is the most commonly used standard in this method. The method has a penalty score of 6 due to the use of low-risk reagents and solvents with a volume of less than 10 mL.

As presented above, the quantification of polyphenols using spectrophotometric techniques requires the absorption of derived compounds using UV–Vis light. Additionally, near-infrared and fluorescence spectroscopy are popular techniques for polyphenol quantification. Near-infrared spectroscopy (NIRS) is a rapid, reliable, and non-destructive technique that does not require sample derivatization or solvent use. Consequently, this technique is considered environmentally friendly, or “green”, compared to other spectrophotometric techniques. NIRS can detect specific functional groups within polyphenols at different wavelengths in the near-infrared range (12,500 to 4000 cm^−1^ or 800 to 2500 nm), including methyl, methylene, and ethylene moieties with the first overtone of O–H and –CH stretching vibration, which are assigned to catechins and epicatechins [79,80]. To analyze samples using NIRS, they must be dehydrated and ground beforehand. The penalty points for NIRS are associated with the sampling preparation and actual testing, such as the energy consumption of the drier, grinder, and NIR equipment. However, since these processes are rapid and have low energy requirements, the penalty points for this technique are low. NIRS is suitable for determining the total procyanidin content in cacao beans, which were tested unaltered and showed a content range of 0.6–19 mg eq. g^−1^ d.w [81]. One of the main limitations of this technique is that a reference set should be generated using alternative techniques such as absorbance spectroscopy or HPLC, and the values of the reference set should be entered in the corresponding software along with the collected NIR spectra for each sample. To generate a reliable reference set, 60 to 100 samples are required, as recommended by the manufacturer. However, the environmental impact of this procedure is small, as it only needs to be performed once for similar samples, after which the established database can be used for years without the need to collect information again.

Similar to NIRS, fluorescence spectroscopy (FS) techniques allow determining polyphenol composition without adding reagents or compound derivatization since they rely on the auto-fluorescence of the polyphenols present in the sample. If the sample requires a dilution step of the extract to avoid inner filter effects, this could contribute to additional PP. Ramirez-Sanchez and co-workers [68] used steady-state fluorescence spectroscopy to detect (−)-epicatechin in microsamples from cacao seeds and cocoa products. They compared the results with the Folin–Ciocalteu (F–C) method and found that the polyphenol quantification was lower than that obtained using the reference method (F–C). However, measuring the autofluorescence of the samples is a fast and reliable green methodology with values of 1 PP due to energy use (Appendix A). Their findings demonstrate the potential of fluorescence spectroscopy for determining polyphenols, specifically (−)-epicatechin, in cacao seeds and cocoa products.

### 5.2. Chromatographic Methods

Spectrophotometric techniques are useful for easy, quick, and economical screening of extracts from cacao and cocoa products. Still, due to the complexity of the cocoa matrix and lack of specificity, the results may not provide a thorough identification of the present compounds. As an alternative, chromatographic techniques coupled to different detectors permit the separation and identification of individual molecules based on their molecular weight, stereo-chemistry, and polarity, constituting a more accurate technique for evaluating polyphenols from cacao and cocoa products. However, some phenols might react unpredictably with spectrophotometric assay reagents, leading to erroneous results, further emphasizing this method’s limitations.

#### Liquid Chromatography

Liquid chromatography (LC) techniques, including high-performance liquid chromatography (HPLC), ultra-HPLC (UHPLC), and ultra-performance liquid chromatography (UPLC), are often chosen for the identification and quantification of polyphenols. The main differences among them are the required time and solvent consumption. UPLC and UHPLC can attain higher pressures than HPLC, resulting in higher flow rates and shorter runs. Liquid chromatography (LC) incurs penalty points due to the energy equipment consumption and the amount and risk of solvents used.

Analyzing polyphenols in cocoa products requires proper identification and quantification, which can be achieved through a wide range of detectors. The multi-wavelength UV–Vis diode array detector (DAD) coupled with an HPLC system is the most popular used [23,35,43,48,54,55,82,83]. Photodiode array detectors (PDA) have also been utilized to detect specific polyphenols in cocoa bean shells, like 5-caffeoylquinic acid and epicatechin [27]. Fluorescence detectors (FD) have also been used to study cacao products [47,49,51,84], but their main drawback is co-elution of peaks due to a lack of detector resolution. Mass spectroscopy (MS) detection systems are used to overcome co-elution and selectivity problems. Out of the several ionization sources [85], electrospray ionization (ESI) is the most popular for analyzing phenolic compounds in cocoa products. The most widely used analyzers for polyphenol detection are quadrupole (Q) and time-of-flight (TOF). Coupling LC equipment with one mass spectrometer, or up to three, allows for detailed analysis of polyphenols [85,86,87], as seen in studies analyzing (−)-epicatechin, (+)-catechin, (−)-catechin, epicatechin, catechin, procyanidin B1, procyanidin B2, procyanidin B5, and procyanidin dimers in unroasted and roasted cocoa beans [4]. The use of MS detectors has its advantages, such as sensitivity and mass range, but it also incurs extra costs and energy consumption, resulting in higher PP.

On the other hand, gas chromatography (GC) can also be used to separate and quantify polyphenols. This technique requires an extra step, i.e., compound derivatization (methylation and acetylation), since it can only measure volatiles. Lack of derivatization led to the unsuccessful detection of polyphenols in cocoa bean shells even after several steps of extraction [63]. This step is time-consuming and involves the use of additional reagents that can impact the sustainability of the method.

UPLC coupled with MS detectors is preferred for analyzing polyphenols in cocoa products due to its shorter analysis time, higher peak efficiency, higher resolution, and higher sensitivity than HPLC. A study comparing the efficiency of HPLC with NP column and UPLC with RP column coupled to a triple quadrupole found that UPLC [24] reduced analysis time by a factor of seven (from 80 to 12.5 min) and allowed the detection and quantification of oligomers from trimer to monomers, which was not possible with HPLC. On the other hand, the environmental impact of chromatographic methods is directly related to the duration of the analysis process, the solvent selection, and the quantity used. The amount of waste generated is proportional to the flow, and longer processes are often used due to the type of equipment or column properties. Therefore, selecting a method with a lower flow and shorter duration can significantly reduce the environmental impact.

Table 3 shows the PPs associated with the identification and quantification of polyphenols based on selected procedures reported in the literature.

In general, UV–Vis spectrophotometric techniques using Folin–Ciocalteau reagents and gallic acid as a standard resulted in 4 PP, which is relatively low. Resourcing to more specific and comprehensive methods, such as the chromatographic technique using HPLC–DAD with methanol, water, acetonitrile, and H_3_PO_4_ as mobile phases, significantly increases the environmental impact (27 PP); robustness and comprehensiveness come at a cost. The selection of the mobile phases for chromatographic techniques, such as HPLC–FLD with water, acetonitrile, and acetic acid, only has a mild effect on the sustainability of the method, obtaining 26 PP. Table 3 shows that all the chromatographic techniques reported tend to have higher penalty points than the spectrophotometric techniques, regardless of their type or operating wavelengths (e.g., NIR, UV–Vis, or fluorescence). This can be attributed to the length of the measurement (minutes for spectroscopy vs. hours for chromatography) and the requirement of reagents other than the standards. Hence, mindfully aiming for the resolution and specificity required during the identification and quantification of polyphenols should be considered when choosing a method, as they can impact the overall cost and sustainability of the analysis.

## 6. Polyphenol Yield Comparison

Table 4 allows for a comparison of the environmental impact and the yield of the technique.

The availability of extraction techniques based on significantly different principles, which also require different levels of pre-treatment, determined that the PP associated with the extraction steps covered a broader range, from 7 to 48, than the analytical steps. The identification and quantification of the polyphenols are often performed by a more limited pool of techniques, which also is reflected in their PP, which ranged from 4–12. Although in some studies, a high penalty point value for extraction corresponded to a high polyphenol yield, this was not always the rule. For instance, the highest yields (i.e., 140 meq g^−1^) were obtained from fermented and sundried nibs, with 36 PP for the extraction. However, a similar yield (about 120 meq g^−1^) could be achieved using extractions with less environmental impact, e.g., 23 PP for fermented and sundried beans. This suggests that the correlation between penalty points and polyphenol yield is not always straightforward and that a mindful selection of extraction conditions and feedstocks can result in high yields. It should be noted that there can be other factors at play, such as the nature of the polyphenols in each material and their accessibility for extraction. Therefore, further studies are required to comprehensively understand the relationship between extraction efficiency and polyphenol yield. Table 4 is intended to guide the reader in understanding the role that extraction and identification might have on the environment to inform future decisions.

## 7. Conclusions

Cacao feedstocks are abundant sources of polyphenols known to possess numerous health benefits. To achieve efficient extraction and accurate identification with minimal environmental impact, careful consideration of pre-treatment, extraction, and analytical determination conditions is required. Pre-treatment methods, such as milling and drying, as well as the use of pulsed electric fields, have been shown to effectively release intracellular compounds, depending on the specific feedstock and targeted analytes. Pre-treatments that are rapid and involve low energy consumption are preferred from a sustainability perspective, while other techniques, such as defatting, which requires solvents and their posterior elimination, have a considerable environmental impact and should not be prioritized. The transition from traditional solid–liquid extraction methods to supercritical fluid extraction is gaining popularity and should be supported due to its high yields and green classification. Using water and only one solvent (ethanol) together with some mechanical pre-treatments can lead to only 10 PP for fermented, sundried, and roasted nibs [77]. Using supercritical fluid extraction with CO_2_ can lead to 12 PP for fermented and sundried cocoa shells [64]. When more than two organic solvents are used, the PP normally range from 17 to 40 (Table 2).

Polyphenol identification and quantification are commonly performed using spectrophotometric methods, with UV–Vis absorbance spectroscopy being widely used. Several assays, including the most popular Folin–Ciocalteu (F–C), can be successfully applied to quantify total polyphenols. Although the penalty points derived from the reagent risks can affect the environmental standing of the method, in general, the penalty points associated with the spectrometric methods are relatively low, ranging from 4 to 12 PP (Table 4).

For more accurate identification of polyphenols in cacao and cocoa products, chromatographic techniques such as HPLC, UHPLC, and UPLC, coupled with different detectors or mass spectrometry, are the preferred methods. The selection of solvents and their separation conditions will play a crucial role in determining the environmental impact of the technique. It is noteworthy that the analysis duration, solvent selection, and the quantity employed carry substantial environmental implications. Consequently, it is imperative to consider the associated penalty points while selecting the appropriate method. The PPs associated with the chromatographic techniques ranged from 12 to 52 (Table 4).

The PP cannot be analyzed on their own, as the polyphenol yield is the goal of the extraction and identification/quantification. In this regard, the higher yields were obtained when methanol was used to extract from cacao husk [43], or a combination of hexane, acetone, and acetic acid to extract from the beans [62] and a combination of hexane, methanol, and acetone to extract from the nibs [54]. Those three methods carried PP of 19, 23, and 36, respectively, with a yield of 151, 40–120, and 140 meq g^−1^, respectively. The highest yield from cacao shells was obtained when a combination of ethanol and water was used in the extraction, giving 19 PPs and a yield of 21–55 meq g^−1^.

Mazzutti et al. [58] showed the lowest PPs in the extraction, as it used a combination of water and ethanol, but it gave a polyphenol yield of 43 meq g^−1^ in the cacao husks. The tables presented herein (Table 2, Table 3 and Table 4 and Appendix A) serve the purpose of consolidating dispersed information found in the literature. This compilation aids readers in the judicious selection of pre-treatment, extraction, and analytical methodologies, contingent upon the unique attributes of the chosen feedstock and the specific analytes of interest. Additionally, these tables provide insight into penalty point considerations and total polyphenol yields. The users are encouraged to deliberately choose extraction and analysis procedures and conditions to attain maximal yield while concurrently mitigating environmental repercussions. The food industry is always looking for new materials that can be used to enhance already existing food products or to develop new ones. Cacao byproducts, such as husks and shells, are unavoidable in the chocolate industry. Polyphenols obtained from parts of the cacao fruit should be viewed as possible food enrichers. We believe that this review can help the industry make educated choices when the time comes to enrich their products.

## Figures and Tables

**Figure 1 foods-13-02397-f001:**
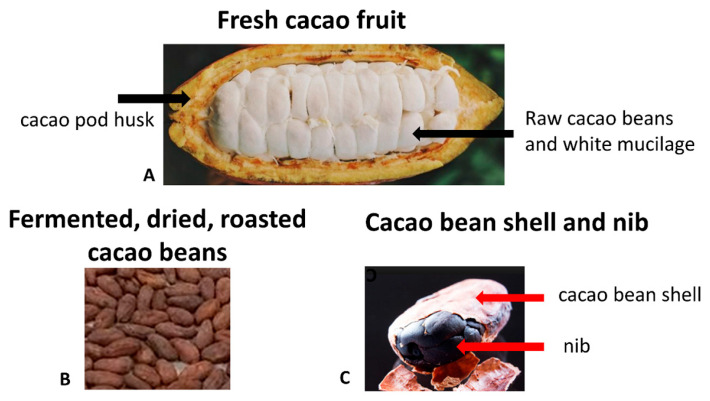
(**A**) Longitudinal cut of a *Theobroma cacao* fruit open in half showing the cacao pod husk, seeds, and mucilage. (**B**) Fermented, sun-dried, and roasted cacao beans. (**C**) Cacao bean shell and nib.

**Figure 2 foods-13-02397-f002:**
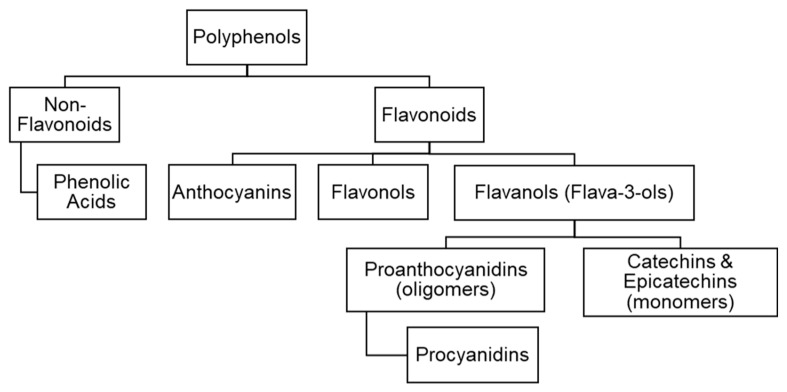
Classification of polyphenols in the *Theobroma cacao* fruit based on their chemical structure.

**Table 1 foods-13-02397-t001:** Eco-Scale penalty points (PPs) assigned to reagents/solvents and equipment used in this review as described by [11] for sample pre-treatment, extraction, and analytical methods.

Criteria				Penalty Points (PP) Assigned
Reagent/Solvent Used and Attributes			Equipment Used	
Amount Used (mL or g)	Risk		Energy Consumption (kWh)	
	Number of Pictograms in Label	Signal Word in Label		
0	0	No signal word	<0.1	**0**
<10	1	“Warning”	0.1–1.5	**1**
10–100	2	“Hazard”	>1.5	**2**
>100	3			**3**

**Table 2 foods-13-02397-t002:** Summary of the pre-treatment and extraction methods for different feedstock from the *Theobroma cacao* L. fruit and the computed penalty points (PPs) obtained by using Appendix A.

Material	State	Extraction Conditions				PP	Ref.
		Pre-Treatment Step		Extraction Step			
		Reagents	Equipment	Reagents/Solvents	Equipment		
Pod Husk	Raw	-	Electric oven, mill	Methanol	Mixer, shaker, rotary evaporator, fridge	19	[43]
		-	Blender, dryer, microwave, freezer	Acetone, acetic acid, water	Freezer, shaker, centrifuge	17	[48]
Beans	Raw	Water, hexane	Freeze-dryer, grinder, dryer	Water, methanol, acetic acid	US bath, shaker, centrifuge	23	[49]
	Fermented; Sundried	Hexane	Incubator, dryer	Methanol, water	Centrifuge, freezer, shaker	21	[52]
		Petroleum ether	-	Water, acetic acid, acetonitrile	Blender, US bath, centrifuge, heater, fridge	25	[36]
		Hexane	Grinder, centrifuge	Acetone, acetic acid, water	US bath, centrifuge, vortex, freezer	23	[53]
	Fermented; Sundried; Roasted	Dry ice, n-hexane	Mill, fridge	Water; DMSO, acetone, diatomaceous earth	PLE	20	[4]
		Hexane, nitrogen	Mill	Acetone; acetic acid; water	US bath, vortex, centrifuge	19	[39]
Nibs	Fermented; Sundried	Hexane	Shaker	Methanol, water, acetone	US bath, centrifuge, rotary evaporator, thermostatic bath	36	[54]
		n-hexane	Mill, centrifuge, freeze-dryer	Acetone, water	Centrifuge	26	[23]
	Fermented; Sundried; Roasted	Petroleum ether	Grinder, centrifuge, fridge, thermostat	Water	Homogenizer, shaking water bath	10	[55]
		Heptane	Centrifuge, roaster	Acetone, acetic acid, water, nitrogen	Centrifuge PHWE, rotary evaporator, US bath, freezer	34	[49]
		Hexane	Centrifuge grinder, shaker	Acetic acid, acetone, water	Rotary evaporator, centrifuge, freeze-dryer	40	[56]
Shells	Raw	-	Freeze-dryer, blender	Ethanol	US bath, rotary evaporator, thermostatic bath	16	[46]
	Fermented; Sundried; Roasted	Hexane	Grinder, orbital shaker	Water, acetic acid, acetone	Rotary evaporator, centrifuge, vortex; freeze-dryer	38	[57]
		Hexane	Mill, Soxhlet, freeze-dryer	Water	Centrifuge	18	[47]
		-	Grinder	CO_2_	SFE, freezer	12	[58]
		-	Grinder	Ethanol	PLE; fridge, rotary evaporator	15	[58]
		-	PEF, mill, grinder, centrifuge, vacuum	Ethanol	Centrifuge, orbital shaker, freezer	18	[29]

**Table 3 foods-13-02397-t003:** Summary of the identification and quantification methods for different feedstock from *Theobroma cacao* L. and the computed penalty points obtained by using Appendix A.

Material	State	Identification and Quantification of Polyphenols				PP	Ref.
		Technique *^1^	Reagents/Solvents *^2^	Standard	Equipment		
Pod Husk	Raw	S	F–C	Gallic acid	UV–Vis spectrometer	4	[43,48]
		C	MeOH, water, acetonitrile, H_3_PO_4_	Gallic, vanillic, caffeic, ferulic, ellagic acid	HPLC–DAD	27	[43]
		C	Formic acid, methanol, water	Gallic acid, catechin, quercetin, epicatechin, p-coumaric A., protocatechuic A	HPLC–DAD	18	[48]
Beans	Raw	S	F–C	Gallic acid	UV–Vis spectrometer	4	[79]
			-	-	NIRS	4	[79]
			F–C	Ferulic acid	UV–Vis spectrometer	5	[88]
		C	-	-	Electronic tongue	4	[79]
	Fermented, Sundried	S	F–C	Gallic acid	UV–Vis spectrometer	4	[52,53]
			F–C	Epicatechin	UV–Vis spectrometer	4	[36]
		C	Water, acetonitrile, acetic acid	Epicatechin, catechin	HPLC–FLD	26	[36]
			Dichloromethane, water, methanol, acetic acid	Epicatechin	HPLC–UV	24	[53]
	Fermented, Sundried, Roasted	S	F–C	Gallic acid	UV–Vis spectrometer	4	[39]
		C	Borate buffer, water, hydroxypropyl-y-cyclodextrin, sodium hydroxide, acetonitrile, formic acid	Epicatechin, catechin	UHPLC–UV–QqQ	24	[4]
			Acetic acid, methanol	Epicatechin, catechin, catechin gallate, gallocatechin, epigallocatechin	Capillary electrophoresis	20	[39]
			Acetic acid, methanol		HPLC–DAD	26	[39]
			Dichloromethane, methanol, water, acetic acid		HPLC–UV–QqQ	25	[39]
Nibs	Fermented, Sundried	S	F–C	Gallic acid	UV–Vis spectrometer	4	[54]
			F–C	Epicatechin	UV–Vis spectrometer	4	[38]
		C	Water, formic acid, ethanol, acetonitrile	Epicatechin, catechin	HPLC–DAD	20	[54]
			Ethyl acetate, butanol, water, 2-propanol, Sephadex LH-20, propanol, acetone	Catechin, epicatechin, PA B2, B3, B4, PA C1, o-arabinoside, cinnamtannin A2, quercetin, quercetin-3-o-glycoside	Semi-preparative SCPC, HPLC–ESI–Q	52	[38]
	Fermented, Sundried, Roasted	S	F–C	Gallic acid	UV–Vis spectrometer	4	[49,55]
		C	Water, formic acid, methanol, acetonitrile	Catechin, epicatechin, procyanidin B1, Procyanidin B2	HPLC–DAD	20	[55]
			Ammonium formate, acetonitrile, methanol, nitrogen, formic acid, acetic acid, sodium formate	Catechin, epicatechin, procyanidin B2	HPLC–DAD–ECD–CAD, HPLC–DAD–MS, HPLC–FLD	42	[49]
Bean Shells	Raw	S	Aluminum chloride	Rutin	UV–Vis spectrometer	11	[46]
		C	Ethanol, water, formic acid, acetonitrile	Procyanidin B2, epicatechin	UPHLC–Q–TOF	22	[46]
	Fermented, Sundried, Roasted	S	F–C	Gallic acid	UV–Vis spectrometer	4	[28,45,58]
			Ethanol, HCl, DMAC	Epicatechin	UV–Vis Spectrometer	12	[57]
		C	Water, methanol, formic acid	Procyanidin B2, epicatechin, catechin, theobromine, caffeine	UPLC–ESI–QqQ	12	[57]
			Helium		GC–MS	4	[58]
			Water, formic acid, methanol	5-caffeoylquinic acid, epicatechin, caffeine, theobromine	HPLC–PDA	18	[28]

*^1^ S = spectrophotometric; C = chromatographic. *^2^ F–C= Folin–Ciocalteau reagents.

**Table 4 foods-13-02397-t004:** Summary of penalty points for different cacao feedstock due to extraction, analysis, and total polyphenol yield measured using non-destructive techniques.

Material	State	PP		Yield	Ref.
		Extraction	Analysis	Total Phenol Content (meq g^−1^)	
Pod Husk	Raw	19	4	151	[43]
		17	4	5–20	[48]
Beans	Raw	7	5	0.08–0.12	[88]
	Fermented, Sundried	21	4	0.6–6	[52]
		25	4	0.2–0.3	[35]
		23	4	40–120	[61]
	Fermented, Sundried, Roasted	16	4	30–70	[39]
Nibs	Fermented, Sundried	36	4	140	[54]
		26	4	80–120	[38]
	Fermented, Sundried, Roasted	34	4	10–35	[49]
Bean Shells	Raw	16	11	7.4	[46]
	Fermented, Sundried, Roasted	48	12	1–4	[57]
		25	4	43	[58]
		18	4	6	[47]
		18	4	21–55	[28]

## Data Availability

No new data were created or analyzed in this study. Data sharing is not applicable to this article.

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
