# Peer review of "Extraction, Identification, and Quantification of Polyphenols from the Theobroma cacao L. Fruit: Yield vs. Environmental Friendliness"

_foods, 2024, doi:10.3390/foods13152397_

Round 1
Reviewer 1 Report
Comments and Suggestions for Authors
This review focuses on the extraction, identification, and quantification of polyphenols from different parts of the cacao fruit, including pods, beans, nibs, and shells. It compares various extraction processes and analytical techniques. The content and composition of polyphenols in cacao depend on factors like tree genotype, growing conditions, and processing methods. As authors recommended, the environmental impact of the extraction and analytical methods is assessed using the Eco-Scale Penalty Point system, emphasizing the importance of selecting sustainable and green chemistry practices. Although this review has a good organization, it also deserves major improvement. The detailed comments are listed as follows.
1. Different extraction protocols are introduced; however, the extraction efficiencies of these methods should be compared in detail. In addition, the optimal method with the favorable extraction efficiency should be indicated clearly in the context.
2. The sources of cacao may affect the actual extracted components. Thus, authors need to classify them into different sources and compare the difference of extracted components.
3. Another key point lies that polyphenols can be also extracted from a vast of plants (RSC Adv., 2021, 11(50): 31693-31711). Authors need to make a brief comparison on their difference. That means the influence of genetic factors on polyphenol content and composition requires further exploration.
4. Importantly, please discuss the yield of polyphenols from different parts of the cocoa fruit in more detail, considering the impact of various extraction methods.
5. The title of this work includes a concept of "sustainability". How to assess the sustainability of the current practices?
Author Response
Overall Response: We want to thank the reviewer for the constructive and thorough assessment of our manuscript. We believe we have addressed all their concerns, and consequently, the manuscript’s quality has improved. Besides addressing the reviewer's suggestions, we have also identified small additional changes that improved the manuscript’s readability. A detailed response to the reviewers’ recommendations is provided below in the order in which the comments were provided. Additionally, the revised manuscript with all modifications marked in red is attached
Comment 1: This review focuses on the extraction, identification, and quantification of polyphenols from different parts of the cacao fruit, including pods, beans, nibs, and shells. It compares various extraction processes and analytical techniques. The content and composition of polyphenols in cacao depend on factors like tree genotype, growing conditions, and processing methods. As authors recommended, the environmental impact of the extraction and analytical methods is assessed using the Eco-Scale Penalty Point system, emphasizing the importance of selecting sustainable and green chemistry practices. Although this review has a good organization, it also deserves major improvement. The detailed comments are listed as follows.
1-Different extraction protocols are introduced; however, the extraction efficiencies of these methods should be compared in detail. In addition, the optimal method with the favorable extraction efficiency should be indicated clearly in the context.
Response 1: Thanks for your encouraging words and thoughtful suggestions. We have extensively revised the manuscript, emphasizing its primary purpose, i.e., calling attention to balancing method performance and its environmental friendliness. The purpose of Section 6 and Table 4 was to draw the comparison that the reviewer alludes to using the extant information in the literature. To better convey this, several changes have been introduced throughout the manuscript.
Lines 83-92 were rephrased to better explain the aim of the review. Currently, they read:
“This review used the Eco-Scale Penalty Point System [13] to compare protocols for the identification and quantification of polyphenolic compounds across different cacao fruit feedstocks. This Penalty Point system was employed to determine the potential impact the method and the analytical techniques have on human health as well as the environment. This review shows that polyphenol yield (i.e., extraction efficacy) is highly dependent on the methods used for extraction, identification, and quantification, which also play a role in the sustainability of a methodology.”
Lines 96-98 were also rephrased to emphasize efficacy as follows:
“Moreover, for readers looking to adopt a suitable scaling-up process, this review should provide valuable insights into the efficacy and long-term viability of industrial extraction methods and monitoring protocols.”
It should be noted that this review also standardized and recalculated all reported yields, which are reported “per g of feedstock” to facilitate comparison (as reported in Table 4)
Comment 2: The sources of cacao may affect the actual extracted components. Thus, authors need to classify them into different sources and compare the difference of extracted components.
Response 2: We agree with the reviewer about the potential impact of the origin of the cacao on polyphenol content and its consequent extraction. We added a brief paragraph early in the introduction, providing some reported examples.
Lines 37-41: “Among the different varieties, Forastero, which is the most widely cultivated cocoa clone worldwide with high cocoa butter content and plain flavor, has been linked to higher levels of polyphenols than Criollo, which has lower anthocyanins and catechins than most varieties [1,2].”
For the purpose of this review it was challenging to find the same extraction/quantification method applied to different feedstocks, which prevented us from making a complete comparison based on origin. In those cases where we found the same techniques used for several feedstocks, we explicitly mentioned them in the text. For example:
Lines 248-265 discuss the PP for the different pre-treatment methods
Lines 288-305 discuss extraction methods on different feedstocks and different authors.
Lines 357-368 introduce Table 2 and discuss PP for different methods, too.
Comment 3: Another key point lies that polyphenols can be also extracted from a vast of plants (RSC Adv., 2021, 11(50): 31693-31711). Authors need to make a brief comparison on their difference. That means the influence of genetic factors on polyphenol content and composition requires further exploration.
Response 3: We also agree with the reviewer that the sources of polyphenols are extensive and numerous and that their genetic makeup will affect polyphenol content and composition. However, these factors in our opinion, are beyond the scope of this review. Herein, we tried to emphasize the importance of the extraction and analytical protocols selected to obtain a reasonable polyphenol yield while aligning with green chemistry principles.
Comment 4: Importantly, please discuss the yield of polyphenols from different parts of the cocoa fruit in more detail, considering the impact of various extraction methods.
Response 4: Thanks for the suggestion and for pointing out this shortcoming. We have revised the manuscript and particularly the conclusions to better summarize the yield and environmental impact of the different methods (also taking into consideration the fruit part involved)
The following paragraphs were added:
Lines 616-620: “Using water and only one solvent (ethanol) together with some mechanical pre-treatments can lead to only 10 PP for fermented, sundried and roasted nibs [73]. But, when more than two organic solvents are used, the PP normally range from 17-40 (Table 2). Conversely, supercritical fluid extraction with CO2 can result in 12 PP for fermented and sundried cocoa shells [58].
Lines 634-646: “The PP associated with the chromatographic techniques ranged from 12 to 52 (Table 4)
Penalty points cannot be the only criterion for selecting a method since obtaining high polyphenol yields is the goal of any extraction, followed by identification/quantification. In this regard, higher yields were obtained when methanol was used as a solvent for cacao husk [43]. Conversely, solvent mixtures were required to achieve high yields from beans and nibs. For example, hexane/acetone/acetic acid and hexane/methanol/acetone mixture resulted in higher yields from the beans [55] and nibs [54], respectively. The listed extractions carried 9, 23 and 36 PP and yields of 151, 40-120 and 140 meq g-1, respectively. The highest yield from cacao shells was obtained when a combination of ethanol/water was used in the extraction, giving 19 PP and a yield of 21-55 meq g-1. The lowest PP reported for extraction used a combination of water and ethanol as solvents, but its yield only reached 43 meq g-1 from cacao husks [58].”
Comment 5: The title of this work includes a concept of “sustainability”. How to assess the sustainability of the current practices?
Response 5: Thanks for the valid and constructive suggestion. In our opinion, the potential sustainability of the extraction and quantification methods is pondered throughout the manuscript using the Eco-Scale Penalty Point system, which covers the consumption of energy due to the number of hours of running the equipment as well as the environmental impact of the chemicals used. Still, we understand the point raised by the reviewer and that the current title might mislead the reader. Hence, we have replaced the title by: Extraction, Identification, and Quantification of Polyphenols from the Theobroma cacao L. fruit: Performance vs. Environmental Friendliness.
Additionally, we revised the manuscript to avoid misinterpretations and better convey the impact of using the Eco Scale to assess the environmental friendliness of the methods.

Reviewer 2 Report
Comments and Suggestions for Authors
The presented review is very interesting to read, offering an overview of the extraction and identification techniques for polyphenols. However, I feel that the title does not emphasize the basic essence of this review - and that is, at least based on my understanding of it, to underline the penalty point system and the environmental friendliness of the extraction and identification processes of polyphenols in cacao. Therefore, I suggest rewriting the title to mention the penalty point system and environmental friendliness of the methods, because that is the main "selling point" of this review.
Also, in specific parts describing the extraction techniques and quantification contain parts that are to general. A sentence or two about the theory of the method is enough, the rest of the text should be focused on specific cocoa applications.
Other comments:
P1 and P2: I think that the graphical abstract should not be a part of the main manuscript file. Please remove it from the main text and upload as a separate file.
P2, keywords: According to the Author instructions, three to ten keywords should be used. You have 11. Also, there are 3 keywords with the words cacao or cocoa, which is unnecessary, since the search algorithms search by word, not by wider meaning. Please leave just one keyword mentioning cacao or cocoa.
P3, L61, L75: please correct the reference format.
P5, L125-126: please correct the reference format. Also, there are a lot of references which are incorrectly formatted throughout the manuscript. Please check the whole text and format the references according to Author instructions.
P5, L154: Equations should be written in an equation format, not as text. Please correct.
Author Response
Overall Response: We want to thank the reviewer for the constructive and thorough assessment of our manuscript. We believe we have addressed all their concerns, and consequently, the manuscript’s quality has improved. Besides addressing the reviewer's suggestions, we have also identified small additional changes that improved the manuscript’s readability. A detailed response to the reviewers’ recommendations is provided below in the order in which the comments were provided. Additionally, the revised manuscript with all modifications marked in red is attached.
Comment 1: The presented review is very interesting to read, offering an overview of the extraction and identification techniques for polyphenols. However, I feel that the title does not emphasize the basic essence of this review - and that is, at least based on my understanding of it, to underline the penalty point system and the environmental friendliness of the extraction and identification processes of polyphenols in cacao. Therefore, I suggest rewriting the title to mention the penalty point system and environmental friendliness of the methods, because that is the main “selling point” of this review.
Response 1: Thanks for your encouraging words. We appreciate the suggestion and modify the title to emphasize the balance between yield and environmental friendliness.
The revised title reads: Extraction, Identification, and Quantification of Polyphenols from the Theobroma cacao L. fruit: Performance vs. Environmental Friendliness.
Comment 2: Also, in specific parts describing the extraction techniques and quantification contain parts that are too general. A sentence or two about the theory of the method is enough, the rest of the text should be focused on specific cocoa applications.
Response 2: Thanks for the suggestion. We revised the manuscript and eliminated those parts in extraction and quantification techniques that were too general. We reduced the paragraphs for PEF (Lines 226-227), PLE (313-314), SFE (323-324), UV (379-384), LC (517-521), MS (532-533).
Comment 3: P1 and P2: I think that the graphical abstract should not be a part of the main manuscript file. Please remove it from the main text and upload as a separate file.
Response 3: The graphical abstract was uploaded as an individual file. We also mistakenly added it to the main text. We have deleted it from the main text in the revised version, following the instructions.
Comment 4: P2, keywords: According to the Author instructions, three to ten keywords should be used. You have 11. Also, there are 3 keywords with the words cacao or cocoa, which is unnecessary, since the search algorithms search by word, not by wider meaning. Please leave just one keyword mentioning cacao or cocoa.
Response 4: Thanks for the suggestions. We have reduced the number of keywords to comply with the journal requirements and eliminated repetitive combinations.
The revised list of keywords is: cacao, cocoa; polyphenols; extraction; analytical methods; green chemistry; sustainability; yield; penalty points.
Comment 5: P3, L61, L75: please correct the reference format.
Response 5: We apologize for the oversight. The reference format was corrected, and the text was revised for instances of the same mistake.
Comment 6: P5, L125-126: please correct the reference format. Also, there are a lot of references which are incorrectly formatted throughout the manuscript. Please check the whole text and format the references according to Author instructions. Mendeley seems not to have done a good job.
Response 6: Thanks for pointing out this problem. As the reviewer kindly inferred, we relied too much on the software. All references were individually (and manually) checked, and changes were enacted when needed.
Comment 7: P5, L154: Equations should be written in an equation format, not as text. Please correct.
Response 7: We agree with the reviewer. The equation was replaced and formatted accordingly.

Reviewer 3 Report
Comments and Suggestions for Authors
In the review publication titled "Extraction, Identification, and Quantification of Polyphenols from the Theobroma cacao L. Fruit: Efficacy and Sustainability," the biological properties of polyphenols found in cocoa fruits (Theobroma cacao L.) were analyzed, as well as the methods for their extraction and qualitative and quantitative identification, considering their significant environmental impact. The currently known information was collected, allowing for the application of the most advantageous methods, starting from the raw material's initial processing, through extraction, to the analysis of the obtained product. I consider the manuscript ready for publication. In the conclusions, I would add a paragraph about the practical applications of extracts and products from cocoa fruits.
Author Response
Overall Response: We want to thank the reviewers for the constructive and thorough assessment of our manuscript. We believe we have addressed all their concerns, and consequently, the manuscript’s quality has improved.
Besides addressing the reviewers’ suggestions, we have also identified small additional changes that improved the manuscript’s readability. A detailed response to the reviewers’ recommendations is provided below in the order in which the comments were provided. Additionally, the revised manuscript with all modifications marked in red is attached.
Comment 1: In the review publication titled “Extraction, Identification, and Quantification of Polyphenols from the Theobroma cacao L. Fruit: Efficacy and Sustainability,” the biological properties of polyphenols found in cocoa fruits (Theobroma cacao L.) were analyzed, as well as the methods for their extraction and qualitative and quantitative identification, considering their significant environmental impact. The currently known information was collected, allowing for the application of the most advantageous methods, starting from the raw material’s initial processing, through extraction, to the analysis of the obtained product. I consider the manuscript ready for publication. In the conclusions, I would add a paragraph about the practical applications of extracts and products from cocoa fruits.
Response 1: Thanks for your encouraging words and suggestion. The following paragraph was added to address the practical applications.
The food industry constantly seeks new ingredients to enhance, improve or replace already existing ones in food products. Cacao husks and shells are unavoidable byproducts in cacao and chocolate manufacturing. Polyphenols obtained from cacao fruit parts should be viewed as potential food-grade bioactive compounds that can improve the sustainability of mainstream cacao products. This review can help the industry decision-making process when manufacturing, disposing of and fortifying their products.

Round 2
Reviewer 1 Report
Comments and Suggestions for Authors
Authors have revised the manuscript properly. It can be accepted.